# Impact of postpartum tenofovir-based antiretroviral therapy on bone mineral density in breastfeeding women with HIV enrolled in a randomized clinical trial

**Lynda Stranix-Chibanda**[1,2]*, **Camlin Tierney**[3], **Dorothy Sebikari**[4], **Jim Aizire**[5], **Sufia Dadabhai**[5], **Admire Zanga**[6], **Cynthia Mukwasi-Kahari**[6], **Tichaona Vhembo**[2], **Avy Violari**[7], **Gerard Theron**[8], **Dhayandre Moodley**[9], **Kathleen George**[10], **Bo Fan**[11], **Markus J. Sommer**[11], **Renee Browning**[12], **Lynne M. Mofenson**[13], **John Shepherd**[11,14], **Bryan Nelson**[3], **Mary Glenn Fowler**[15‡], **George K. Siberry**[16‡], for the PROMISE P1084s study team[¶]

1 Department of Paediatrics and Child Health, Faculty of Medicine and Health Sciences, University of Zimbabwe, Harare, Zimbabwe, 2 University of Zimbabwe Clinical Trials Research Centre, Harare, Zimbabwe, 3 Department of Biostatistics, Center for Biostatistics in AIDS Research, Harvard T.H. Chan School of Public Health, Boston, MA, United States of America, 4 Johns Hopkins University Research Collaboration, Makerere University, Kampala, Uganda, 5 Department of Epidemiology, Johns Hopkins Bloomberg School of Public Health, Baltimore, MA, United States of America, 6 Radiology Department, University of Zimbabwe Faculty of Medicine and Health Sciences, Harare, Zimbabwe, 7 Perinatal HIV Research Unit, Johannesburg, South Africa, 8 Stellenbosch University, Cape Town, South Africa, 9 Centre Aids Prevention Research South Africa (CAPRISA), University of KwaZulu-Natal, Durban, South Africa, 10 FHI 360, IMPAACT Operations Center, Durham, NC, United States of America, 11 Radiology and Biomedical Imaging Unit, University of California San Francisco, San Francisco, CA, United States of America, 12 Division of AIDS, National Institute of Allergy and Infectious Diseases, Bethesda, MA, United States of America, 13 Research Department, Elizabeth Glaser Pediatric AIDS Foundation, Washington, DC, United States of America, 14 University of Hawaii Cancer Center, Honolulu, Hawaii, 15 Department of Pathology, Johns Hopkins University School of Medicine, Baltimore, MA, United States of America, 16 United States Agency for International Development, Arlington, VA, United States of America

‡ These authors are joint senior authors on this work.
¶ Membership of the PROMISE P1084s study team is provided in the Acknowledgments.
* lstranix@uz-ctrc.org

**Data Availability Statement:** The individual level data cannot be made publicly available due to the ethical restrictions in the study's informed consent

## Abstract

### Objectives

*We set out to evaluate the* effect of postnatal exposure to tenofovir-containing antiretroviral therapy on bone mineral density among breastfeeding women living with HIV.

### Design

IMPAACT P1084s is a sub-study of the PROMISE randomized trial conducted in four African countries (ClinicalTrials.gov number NCT01066858).

### Methods

IMPAACT P1084s enrolled eligible mother-infant pairs previously randomised in the PROMISE trial at one week after delivery to receive either maternal antiretroviral therapy

documents and in the International Maternal Pediatric Adolescent AIDS Clinical Trials (IMPAACT) Network's approved human subjects protection plan; public availability may compromise participant confidentiality. However, data are available to all interested researchers upon request to the IMPAACT Statistical and Data Management Centre's data access committee by email to sdac. data@fstrf.org or sdac.data@sdac.harvard.edu. This committee reviews and responds to requests for data, obtains necessary approvals from IMPAACT leadership and the NIH, arranges for signature of a Data Use Agreement, and releases the requested data.

**Funding:** Overall support for the International Maternal Pediatric Adolescent AIDS Clinical Trials Network (IMPAACT) was provided by the National Institute of Allergy and Infectious Diseases (NIAID) of the National Institutes of Health (NIH) under Award Numbers UM1AI068632 (IMPAACT Leadership and Operations Center), UM1AI068616 (IMPAACT Statistical and Data Management Center) and UM1AI106716 (IMPAACT Laboratory Center), with co-funding from the Eunice Kennedy Shriver National Institute of Child Health and Human Development (NICHD) and the National Institute of Mental Health (NIMH). The content is solely the responsibility of the authors and does not necessarily represent the official views of the NIH. Antiretrovirals were provided free of charge for the PROMISE P1084s study by AbbVie, Gilead Sciences, and GlaxoSmithKline. Bone density scanners were provided for the PROMISE P1084s study by Gilead Sciences. The funders had no role in study design, data collection and analysis, decision to publish, or preparation of the manuscript.

**Competing interests:** The authors have declared that no competing interests exist.

(Tenofovir disoproxil fumarate / Emtricitabine + Lopinavir/ritonavir–maternal TDF-ART) or administer infant nevirapine, with no maternal antiretroviral therapy, to prevent breastmilk HIV transmission. Maternal lumbar spine and hip bone mineral density were measured using dual-energy x-ray absorptiometry (DXA) at postpartum weeks 1 and 74. We studied the effect of the postpartum randomization on percent change in maternal bone mineral density in an intention-to-treat analysis with a t-test; mean and 95% confidence interval (95%CI) are presented.

## Results

Among 398/400 women included in this analysis, baseline age, body-mass index, CD4 count, mean bone mineral density and alcohol use were comparable between study arms. On average, maternal lumbar spine bone mineral density declined significantly through week 74 in the maternal TDF-ART compared to the infant nevirapine arm; mean difference (95%CI) -2.86 (-4.03, -1.70) percentage points (p-value <0.001). Similarly, maternal hip bone mineral density declined significantly more through week 74 in the maternal TDF-ART compared to the infant nevirapine arm; mean difference -2.29% (-3.20, -1.39) (p-value <0.001). Adjusting for covariates did not change the treatment effect.

## Conclusions

Bone mineral density decline through week 74 postpartum was greater among breastfeeding HIV-infected women randomized to receive maternal TDF-ART during breastfeeding compared to those mothers whose infants received nevirapine prophylaxis.

## Introduction

Bone mineral density (BMD) peaks in early adulthood after which it is lost gradually at a rate of approximately 1% per year [1, 2]. Low BMD, either osteopenia (BMD Z- or T-scores between -1 to -2.5) or osteoporosis (scores <-2.5), may result in low-trauma bone fractures; and women are at increased risk for osteoporosis or osteopenia compared to men. Lifestyle factors that increase this risk include lack of exercise, cigarette smoking, high alcohol intake, and low intake of dietary calcium and vitamin D.

Some of these traditional lifestyle factors for low BMD occur in people living with HIV [3–6]. However, both HIV infection itself and antiretroviral therapy (ART) are also associated with an increased risk of low BMD [5, 7–13], and an accelerated BMD decline is seen in the first year following immediate compared to deferred initiation of ART [12]. Tenofovir disoproxil fumarate (TDF) was implicated to have a relatively greater adverse impact on BMD when compared to other antiretroviral drugs [14], although reduced BMD has been observed with other agents [15, 16]. Low BMD was reported in children, adolescents and adults on tenofovir-containing ART (TDF-ART) [17] with possible improvement after regimen switch in adults [18–20]. Significant BMD decline was also seen in adults without HIV receiving TDF-based pre-exposure prophylaxis (PrEP) [21, 22] and treatment for chronic hepatitis B infection [23, 24]. BMD loss with PrEP use occurred within months of initiation and then stabilized, returning to baseline levels upon removal of the drug [21].

Female sex appears to be independently associated with low BMD among adults with HIV on ART [25]. In women, BMD is also affected by reproductive hormones; there is a well-

documented transient loss during pregnancy and lactation [1, 2, 26–28]. Despite the involvement of regulatory homeostatic pathways, BMD decreases by about 3% during pregnancy [1, 2]. During the initial six months of lactation, there is a further 5%–6% decrease which generally resolves to pre-pregnancy levels within three months following cessation of breastfeeding [29–31]. The rate of bone lost each year in pre-menopausal women enrolled in the US-based Women's Interagency HIV Study (WIHS) was approximately 0.4–0.8% and was not associated with HIV infection status. However, long-term study of the aging WIHS cohort revealed that women living with HIV bear a higher 10-year fracture incidence compared to uninfected women (adjusted Hazard Ratio: 1.32, 95%CI: 1.04, 1.69) [32].

Breastfeeding beyond 12 months is standard for women in Sub-Saharan Africa, regardless of HIV status. Potential effects of prolonged lactation and high parity on BMD and fracture risk are not resolved [2, 27, 33–35]. Most populations studied to date were from developed countries where parity is not high, breastfeeding is not prolonged and nutrition is generally adequate [36]. Nabwire *et al* observed accentuated bone loss after 14 weeks of breastfeeding in Ugandan women with HIV on life-long ART compared to women without HIV [37]. The randomized design of the PROMISE study provided an ideal opportunity to further contribute towards clarifying the relative role of ART in the observed bone loss among women living with HIV. PROMISE participants were women with HIV infection and high CD4 counts in Sub-Saharan Africa who did not meet country criteria for ART initiation at the time of the postpartum randomization to take TDF-ART or no ART while breastfeeding. The IMPAACT P1084s Bone and Kidney Health sub-study of PROMISE aimed to evaluate in a breastfeeding population the effect of postpartum exposure to TDF-ART on change in maternal BMD from delivery to 18 months, when BMD restitution following cessation was expected to have taken place. The full protocol is available on the Network website https://www.impaactnetwork.org/studies/p1084s

## Materials and methods

### Study population

The PROMISE trial enrolled 3747 pregnant women living with HIV along with their infants to determine the optimal antiretroviral strategy to prevent perinatal and postpartum transmission of HIV from mother to child and preserve maternal health and infant survival in 15 countries across different health settings [38–41]. The PROMISE Antepartum randomization was to open-label Zidovudine mono-drug prophylaxis or Zidovudine/Lamivudine/Lopinavir/ritonavir ART (see **S1 Fig**–PROMISE study design).

The PROMISE Postpartum Component randomized healthy women with HIV and high CD4 counts intending to breast feed and their uninfected, healthy infants weighing at least 2kg one week after delivery to receive either open-label maternal ART (Tenofovir disoproxil fumarate/Emtricitabine + Lopinavir/ritonavir–'TDF-ART') or administer infant nevirapine prophylaxis without maternal ART (iNVP) throughout the period of breastfeeding to prevent breastmilk transmission. At the time PROMISE was conducted, enrolled women did not meet the criteria to initiate ART for their own health and life-long ART was not yet standard for pregnant women. The Bone and Kidney Health sub-study offered postpartum co-enrolment to a sub-set of women enrolled in PROMISE with no prior TDF exposure during pregnancy in four African countries with capacity for BMD evaluation; Malawi, South Africa, Uganda and Zimbabwe; with a target sample size of 400.

On July 6, 2015 PROMISE sites were notified that all participants should be offered ART based on the results of the Strategic Timing of AntiRetroviral Treatment (START) study [42] which demonstrated a significant benefit to beginning ART, including in patients with high

CD4 counts. Analyses for P1084s are thus based on data collected at visits through the date of notification.

## Study procedures

After obtaining maternal consent, P1084s sub-study entry occurred immediately (same day) after PROMISE Postpartum randomization on postpartum day 6–14. Participants were followed at 6, 26 and 74 postpartum weeks. Data collected throughout follow-up included socio-demographic information, medical history, HIV-related medical information including viral load, ART adherence assessment, smoking status, alcohol intake status, physical activity level, dietary intake, renal function, concomitant medications including contraceptive use and breastfeeding status. The protocol did not specify nutritional supplements. Weight and height measurements were conducted by trained study staff according to standardized measurement guidance. Maternal participants randomized to iNVP (no ART) who subsequently met immunological or clinical criteria to initiate ART for their own health were immediately started on ART and remained in observational follow-up.

## Bone mineral density assessment

Maternal BMD was measured at the lumbar spine and hip by dual-energy x-ray absorptiometry (DXA). The baseline measurement was scheduled at postpartum week one (day 5–21) and repeated at postpartum week 74 (+/-6 weeks), unless the participant was pregnant. Standardized procedures for obtaining the scan were followed to minimize differences between the study sites–all scanners were Hologic models that were cross-calibrated with a phantom, each technician underwent webinar training and quality review of their first scan, and DXA scans were centrally analyzed at the University of California San Francisco Department of Radiology and Biomedical Imaging. DXA operators and readers were blinded to study treatment assignment.

## Study oversight

The study was funded by the National Institutes of Health (ClinicalTrials.gov number NCT01066858). Written informed consent was obtained from each sub-study participant. Study conduct adhered to international guidelines, and the sub-study was approved by an institutional review board or ethics committee at each site and corresponding collaborating institutions in the United States.

Ethics committees and institutional review boards that approved this study include—MUJHU/Kampala, Uganda: The Joint Clinical Research Centre (JCRC) IRB, the National Drug Authority in Uganda and the Johns Hopkins Medical Institutions (JHMI) IRB in the U. S.; Wits RHI Shandukani CRS and Soweto IMPAACT CRS, Johannesburg, South Africa: University of Witwatersrand Human Ethics Research Committee (Medical), Medicines Control Council (South African Health Products Regulatory Authority in February 2018); FAM-CRU CRS, Cape town, South Africa: Health Research Ethics Committee (HREC), Faculty of Health Sciences, Stellenbosch University and Medicines Control Council (South African Health Products Regulatory Authority in February 2018); Durban Paediatric HIV CRS, Durban, South Africa: University of KwaZulu-Natal (UKZN) Biomedical Research Ethics Committee, Medicines Control Council (South African Health Products Regulatory Authority in February 2018);George CRS, Lusaka, Zambia: University of North Carolina (UNC) at Chapel Hill Biomedical IRB and University of Zambia Biomedical Research Ethics Committee (UNZABREC); Harare, Seke North and St. Mary's sites, Zimbabwe: Medical Research Council of Zimbabwe (MRCZ), Research Council of Zimbabwe (RCZ), Medicine Control Authority of Zimbabwe

(MCAZ), Joint Parirenyatwa group of Hospitals/University of Zimbabwe College of Health Sciences Research Ethics Committee(JREC); Byramjee Jeejeebhoy Medical College (BJMC) CRS, Pune, India: BJ Government College CTU Ethics Committee and Johns Hopkins IRB; Blantyre, Malawi: College of Medicine Research and Ethics Committee (COMREC) in Malawi, Pharmacy, Medicines and Poisons Board and Johns Hopkins Medical Institutions (JHMI) IRB in the U.S.; Lilongwe, Malawi: National Health Sciences Research Committee (NHSRC) in Malawi Pharmacy, Medicines and Poisons Board, and University of North Carolina, Chapel Hill (UNC-CH) Office of Human Research Ethics IRB in the U.S and Kilimanjaro Christian Medical Centre (KCMC), Moshi, Tanzania: Kilimanjaro Christian Medical College Ethics Committee, National Health Research Ethics Committee and Tanzania Medicines and Medical Devices Authority.

## Statistical analysis

The primary outcome measure was percent change in lumbar spine (LS) BMD assessed by DXA between baseline and week 74. A secondary outcome measure was percent change in hip BMD from baseline to week 74. Primary analyses were carried out by randomized assignment for the mothers included in the analysis, as were secondary analyses on additional DXA outcome measures. Selected subgroup and restricted/as-treated analyses were performed as secondary analyses. Percent change in maternal LS BMD and hip BMD were analyzed with Student t-tests (two-sided) to compare the maternal TDF-ART and iNVP arms. The TDF-ART versus iNVP effect was adjusted for covariates and effect modification by subgroups (interaction tests) was assessed via linear regression. Covariates included at postpartum randomization (baseline): age, weight, parity at PROMISE entry, HIV RNA level, country, and antepartum randomization assignment. Additional comparisons applied Wilcoxon/Kruskal-Wallis test for continuous data, and $\chi^2$/exact tests for categorical data, as appropriate. Analyses were performed using SAS v9.4 (Cary, NC). Statistical significance was set at the 0.05 level, with no adjustment for multiple comparisons.

The data cannot be made publicly available due to the ethical restrictions in the study's informed consent documents and in the International Maternal Pediatric Adolescent AIDS Clinical Trials (IMPAACT) Network's approved human subjects protection plan; public availability may compromise participant confidentiality. However, data are available to all interested researchers upon request to the IMPAACT Statistical and Data Management Centre's data access committee by email to sdac.data@fstrf.org or sdac.data@harvard.edu. This committee reviews and responds to requests for data, obtains necessary approvals from IMPAACT leadership and the NIH, arranges for signature of a Data Use Agreement, and releases the requested data.

## Results

### Study participants

Of 2431 mother-infant pairs randomized into the PROMISE Postpartum Component, 592 without prior TDF exposure were randomized at the four participating sites during the sub-study enrolment period. Four hundred mother-infant pairs (202 and 198 pairs previously randomized to TDF-ART and iNVP, respectively) co-enrolled (68%) into the postpartum P1084s sub-study between August 2011 and October 2012 (**Fig 1**).

There were no eligibility violations. Per the P1084s protocol and analysis plan, the primary analysis excluded two participants in the TDF-ART arm whose initial ART regimen after postpartum randomization was not a TDF-containing regimen. Among the 398 mothers included in the analysis, 374 (94%) completed the P1084s study follow-up. Twenty-two (6%) mothers

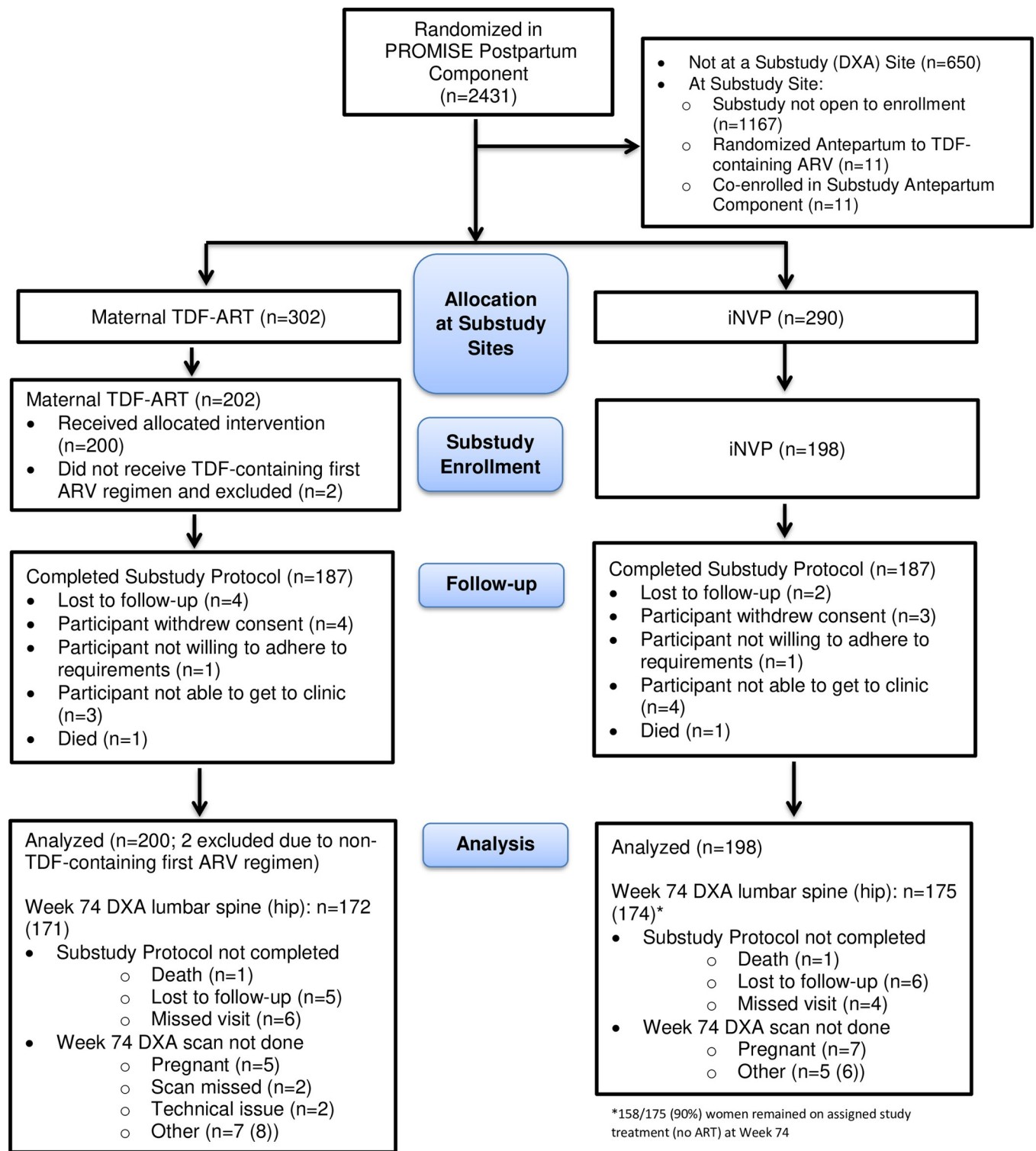

**Fig 1. CONSORT flow diagram.**

discontinued the P1084s sub-study before completing the week 74 visit with 12 (6%) from the maternal TDF-ART arm and 10 (5%) from the iNVP arm, with no apparent differences in

reasons for discontinuation. There was one maternal death in each arm. One hundred and fifty-eight women (90% of those with week 74 lumbar spine DXA scan) in the iNVP arm remained on their assigned study treatment (no ART) at the time of the week 74 DXA and 17/175 women had begun ART prior to that date (primarily TDF-containing ART regimens). Percent change in BMD from baseline to week 74 was available for lumbar spine (hip) in 337 (336)/398 mothers who had DXA scan data at both time points. They were missing on sixty-two (16%) mothers for reasons of prior study follow-up discontinuation (3.3%), missed week 74 visit (2.5%), DXA scan technical problems (1.0%), and not done due to pregnancy (3.0%),

Mothers were enrolled in Uganda (42%), Zimbabwe (42%), South Africa (10%) and Malawi (7%). The women were co-enrolled after delivery of their first (16%), second (29%), third (26%) or fourth-ninth (29%) pregnancies and all had initiated breastfeeding. Baseline characteristics at sub-study entry for those included in the analysis are presented in **Table 1** and were comparable between the study arms (see **S1 Table** for characteristics of those not co-enrolled). Alcohol use in the 30 days prior to sub-study entry was reported by 12% of participants.

The proportion of sub-study participants who had no antenatal HIV intervention (late presenters to care) was 9%. The median CD4 count in pregnancy at initial PROMISE study entry was 540 cells/mm$^3$ (446–701) with 40% of participants having a CD4 count 350 to <500 cells/mm$^3$. At entry into the sub-study postpartum, 46% had been randomized Antepartum to triple ARV with Zidovudine/Lamivudine/Lopinavir/ritonavir ART, the median CD4 count was 671 cells/mm$^3$ (544.0–855) with CD4 350 to <500 cells/mm$^3$ in 18%. The median HIV-1 viral load was 400 copies/ml (61.5–1314.5) in the maternal TDF-ART arm and 400 copies/ml (142–3182) in the iNVP arm.

During follow-up, use of DMPA injectable contraceptive and duration of breastfeeding were similar between study arms. Two thirds (65%) of mothers started contraceptives and one third had any exposure to injectable contraceptives prior to their week 74 DXA measurement. Half (54%) started their first contraceptive at least 12 months prior to their week 74 DXA scan. Median time to cessation of breastfeeding was 61.4 weeks and 48% of the sample had stopped breastfeeding more than three months before the week 74 DXA scan; at DXA scan for lumbar spine the median (25$^{th}$, 75$^{th}$ percentile) number of weeks since weaning was 17.29 (9.07–24.71) in the maternal TDF-ART arm and 17.00 (9.86–23.36) in the iNVP arm. By the time data were censored at release of the START study results in June 2015, there had been one traumatic fracture recorded in the maternal TDF-ART arm and no fractures in the iNVP arm after delivery.

## Change in bone mineral density

The baseline (week 1) mean BMD at the lumbar spine in the maternal TDF-ART study arm was 0.94 g/cm$^2$ (95%CI 0.93–0.96) and 0.94 g/cm$^2$ (0.92–0.95) in the iNVP arm. At the hip, the baseline mean BMD in the maternal TDF-ART study arm was 0.98 g/cm$^2$ (0.96–0.99) and 0.98 g/cm$^2$ (0.96–1.00) in the iNVP arm.

For the primary outcome measure of percent change in LS BMD (**Fig 2**), the maternal TDF-ART arm had a greater mean percent change from baseline to week 74 postpartum of -2.05% (95% CI -2.88, -1.22) compared to a mean of +0.81% (-0.01, 1.64) in the iNVP arm, for a mean difference of -2.86% (-4.03, -1.70) (p-value <0.001). Similarly, for the secondary outcome measure of hip BMD, the mean percent change was greater in the maternal TDF-ART arm compared to the iNVP arm; -5.34% (-5.96, -4.73) versus -3.05% (-3.72, -2.38) respectively, for a mean difference of -2.29% (-3.20, -1.39) (p-value<0.001).

These treatment effects remained after adjusting for the covariates under consideration, either one at a time (results not shown) or together in a model (p<0.001) (**Fig 3**). The treatment difference on lumbar spine BMD was larger for those women who were also randomized

**Table 1. Baseline characteristics.**

| | | Postpartum Randomization Arm | | | |
|---|---|---|---|---|---|
| | | Maternal TDF-ART (N = 200) | iNVP (N = 198) | Total (N = 398) | P-Value |
| Age (years) | Median (Q1-Q3) | 26.3 (23.4–29.5) | 26.7 (23.2–31.3) | 26.5 (23.3–30.2) | 0.73[a] |
| Country | Malawi | 12 (6%) | 15 (8%) | 27 (7%) | 0.34[b] |
| | South Africa | 24 (12%) | 14 (7%) | 38 (10%) | |
| | Uganda | 85 (43%) | 82 (41%) | 167 (42%) | |
| | Zimbabwe | 79 (40%) | 87 (44%) | 166 (42%) | |
| Previous PROMISE Component | 1077BA–breastfeeding | 180 (90%) | 180 (91%) | 360 (90%) | 0.60[b] |
| | 1077BL–late presenters | 19 (10%) | 18 (9%) | 37 (9%) | |
| | 1077FA–formula feeding | 1 (1%) | 0 (0%) | 1 (0%) | |
| PROMISE Antepartum Randomization | # missing (late presenters) | 19 | 18 | 37 | |
| | Triple ARV (3TC-ZDV/LPV-RTV) | 87 (48%) | 98 (54%) | 185 (51%) | 0.22[b] |
| | ZDV+sdNVP+TRV tail | 94 (52%) | 82 (46%) | 176 (49%) | |
| Weight (kg) | Median (Q1-Q3) | 60.6 (55.1–70.3) | 62.0 (56.0–69.6) | 61.2 (55.7–70.0) | 0.74[a] |
| Height (cm) | Median (Q1-Q3) | 158.3 (152.7–163.0) | 159 (152–163) | 159.0 (152.5–163.0) | 0.94[a] |
| BMI (kg/m$^2$) | Median (Q1-Q3) | 24.6 (22.2–28.1) | 24.9 (22.6–27.7) | 24.8 (22.4–28.0) | 0.65[a] |
| CD4 at Antepartum Screening (cells/mm$^3$) | # missing (late presenters) | 19 | 18 | 37 | |
| | Median (Q1-Q3) | 544 (450–707) | 539.0 (434.5–691.5) | 540 (446–701) | 0.31[a] |
| CD4 at Postpartum Screening (cells/mm$^3$)[c] | Median (Q1-Q3) | 683.5 (548.0–873.0) | 665 (542–849) | 671 (544–855) | 0.41[a] |
| HIV RNA level prior to randomization (copies/ml) | Median (Q1-Q3) | 400.0 (61.5–1314.5) | 400 (142–3182) | 400 (83–2289) | 0.07[a] |
| Calcium (mg/dL) | # missing | 10 | 9 | 19 | 0.73[a] |
| | Median (Q1-Q3) | 8.5 (8.2–8.9) | 8.6 (8.1–9.0) | 8.5 (8.2–8.9) | |
| Phosphate (mg/dL) | # missing | 8 | 8 | 16 | 0.97[a] |
| | Median (Q1-Q3) | 3.6 (3.2–4.1) | 3.6 (3.3–4.0) | 3.6 (3.2–4.0) | |
| WHO Stage[c] | # missing | 169 | 167 | 336 | |
| | Clinical stage I | 30 (97%) | 27 (87%) | 57 (92%) | 0.16[b] |
| Number of years of smoking[c] | # missing | 142 | 137 | 279 | |
| | No history of smoking | 57 (98%) | 61 (100%) | 118 (99%) | |
| History of alcohol use prior to randomization[c] | # missing | 162 | 160 | 322 | |
| | Yes | 6 (16%) | 3 (8%) | 9 (12%) | 0.29[b] |
| Week 1 DXA Lumbar Spine (g/cm$^2$) | # missing | 6 | 9 | 15 | |
| | Median (Q1-Q3) | 0.95 (0.87, 1.01) | 0.93 (0.87, 1.00) | 0.94 (0.87, 1.00) | |
| | Mean (95%CI) | 0.94 (0.93–0.96) | 0.94 (0.92–0.95) | 0.94 (0.93–0.95) | |
| | Standard Dev. | 0.11 | 0.11 | 0.11 | |
| Week 1 DXA Hip (g/cm$^2$) | # missing | 4 | 10 | 14 | |
| | Median (Q1-Q3) | 0.98 (0.90, 1.04) | 0.98 (0.90, 1.05) | 0.98 (0.90, 1.05) | |
| | Mean (95%CI) | 0.98 (0.96–0.99) | 0.98 (0.96–1.00) | 0.98 (0.97–0.99) | |
| | Standard Dev. | 0.11 | 0.12 | 0.11 | |

[a]Wilcoxon Test;

[b]Chi-Square Test;

[c]Characteristics with small sample sizes (for WHO Stage, current and history of smoking, and alcohol use) may have had an evaluation >30 days before randomization. Data for Postpartum Screening CD4 Count were missing for one participant (from iNVP arm).

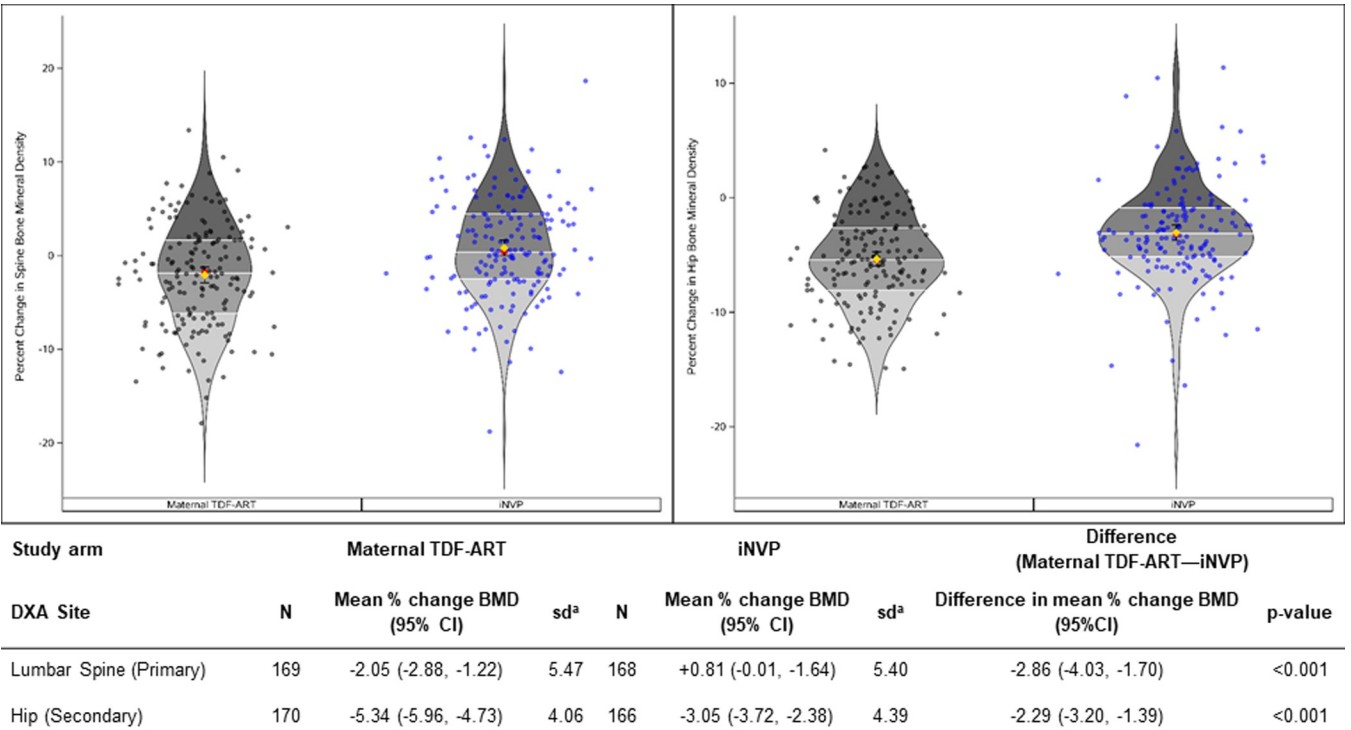

| Study arm | | **Maternal TDF-ART** | | | **iNVP** | | | **Difference (Maternal TDF-ART—iNVP)** | |
| --- | --- | --- | --- | --- | --- | --- | --- | --- | --- |
| DXA Site | N | Mean % change BMD (95% CI) | sd[a] | N | Mean % change BMD (95% CI) | sd[a] | Difference in mean % change BMD (95%CI) | p-value |
| Lumbar Spine (Primary) | 169 | -2.05 (-2.88, -1.22) | 5.47 | 168 | +0.81 (-0.01, -1.64) | 5.40 | -2.86 (-4.03, -1.70) | <0.001 |
| Hip (Secondary) | 170 | -5.34 (-5.96, -4.73) | 4.06 | 166 | -3.05 (-3.72, -2.38) | 4.39 | -2.29 (-3.20, -1.39) | <0.001 |

[a]sd=standard deviation

**Fig 2. Percent change in bone mineral density from postpartum week 1 to week 74 at spine and hip.** Data points and the distributions are summarized with violin plots. Each violin plot presents density, along with horizontal lines indicating quartiles, a red circle for the median and a diamond for the mean (with whiskers for the 95% CI). Individual data points are superimposed on the violin plot.

to ART in the PROMISE antepartum component (p = 0.006). At both lumbar spine and hip the treatment difference was larger for women who weighed less at baseline (p-value< = 0.05). For the secondary outcome measure of hip BMD the treatment difference was larger for younger women (p = 0.03). The treatment effect did not appear to differ by country, breastfeeding or contraception status at either location (p≥0.55). The mean difference in percent BMD change at the lumbar spine in Malawi did appear to be smaller than that seen in other countries for the primary endpoint (-0.70) although the 95%CI was wide (-5.63, 4.23). The effect of

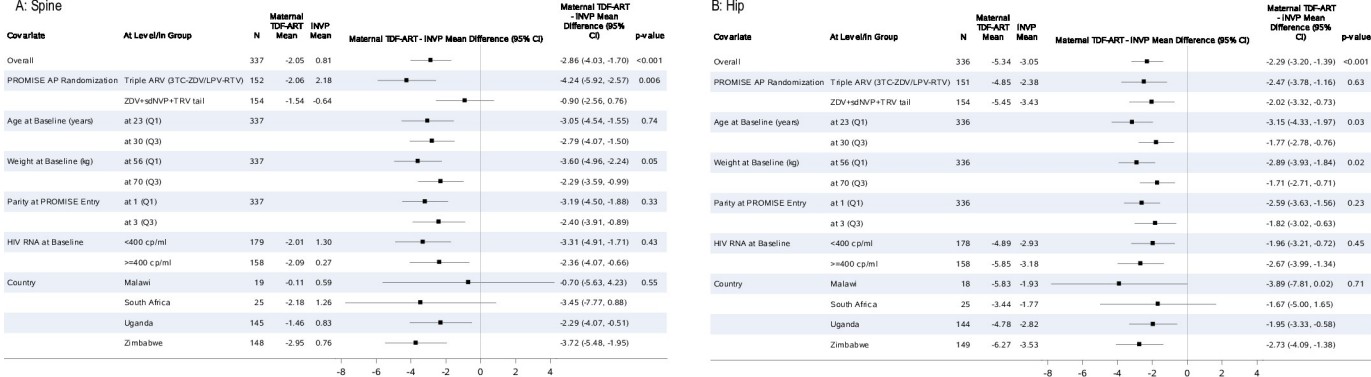

**Fig 3.** Percent change to week 74 in (A) spine BMD (primary outcome measure) and (B) hip BMD: maternal TDF-ART vs. iNVP treatment effect modification by covariate.

ART exposure on BMD at the lumbar spine and hip remained in sensitivity analyses, among women who were still breast feeding or on their assigned treatment at the time of the week 74 DXA scan.

Among 264 participants who had ceased breast feeding before week 74, the mean percent change in LS BMD was -1.39 in the mART arm and -1.31 in the iNVP arm (mean difference (95%CI) -2.71 (-4.00, -1.41). Among the 236 participants who were on their assigned treatment at week 74 (on TDF in the maternal arm, no ARVs in the iNVP arm) the mean percent change in LS BMD was -3.73 in the maternal TDF- and 1.11 in the iNVP arm (mean difference (95% CI) -4.84 (-6.25, -3.43)).

## Discussion

Within a large randomized PMTCT clinical trial among asymptomatic African mothers with HIV and high CD4 counts who breastfed on average for longer than one year, we found mothers randomized postpartum to TDF-ART BMD during breastfeeding experienced a greater BMD decline at the lumbar spine between postpartum weeks 1 and 74 when compared to women who did not receive maternal ART. Our primary analysis included randomized participants who signed up for the substudy. Selection, confounding bias and bias due to drop-out appeared to be small. The overall estimated treatment difference includes participants who subsequently changed their initially assigned regimen and appears to result in a conservative estimate of the difference. The overall effect of TDF-ART on maternal BMD remained in adjusted and sensitivity analyses.

The negative effect on BMD in women with HIV on TDF-ART and on no ART persisted after the period of lactation had ended, when post-weaning restitution/recovery would have been anticipated. Nabwire *et al* [37] also reported partial skeletal recovery with a 2–3% deficit in total hip BMD three months following cessation of breastfeeding among Ugandan women with HIV on Efavirenz-based TDF-ART compared to women living without HIV. Taken together with our findings, bone loss during pregnancy and breastfeeding is potentially exacerbated by TDF-ART. The long-term clinical implication of a 2–3% greater decline in postpartum maternal BMD over 74 weeks in women on TDF-ART is unknown and is smaller than the assumed 5–7% considered of clinical concern in the study design. However, for women with HIV in settings where fertility is high, breastfeeding into the second year post delivery is the norm and where repeated cycles of bone loss/restoration with each pregnancy occur— sometimes at close intervals—the findings raise potential concerns of increased risk for long term bone loss among high parity women on life-long ART. In addition, women in such settings often have additional risk factors that impact bone health, such as undernutrition and predominance of injectable hormonal contraception [43].

The larger treatment difference on lumbar spine BMD observed in the sub-study women who had been randomly assigned to antepartum ART could reflect a cumulative effect of the ARVs they received while pregnant and during lactation on postpartum bone health. While the specific PI-based ART regimen administered in PROMISE is not widely used in the current era of universal ART, Tenofovir based ART regimens are widely used. Tenofovir exposure increased 20–37% when TDF was co-administered with ritonavir (r), Atazanavir/r, Lopinavir/r, Darunavir/r, or Cobicistat [44]. In the PROMISE trial, the postpartum regimen provided was TDF combined with Emtricitabine and Lopinavir/r. The increased TDF exposure with Lopinavir/r might have led to an increased risk of bone toxicity in the context of this study. Investigations continue to evaluate novel ARVs reported to have less effect on BMD, such as Tenofovir alafenamide [45] with the view to optimize the safety and efficacy of ART regimens among women of reproductive potential. Our findings relating to the decreased BMD findings post cessation of

breastfeeding for those HIV Infected women on TDF-ART remain important, as global HIV programs will continue to use TDF in the near future for HIV treatment and prevention (within PrEP regimens) until alternative regimens have demonstrated favorable safety in women and in pregnancy; and given that use of ART in general has been associated with bone loss.

## Relative strengths and limitations

Major strengths of this study design include the randomized trial design, large sample size and inclusion of breastfeeding women living with HIV from multiple sites in Africa. The large sample permitted the detection of small differences between the study arms which is important when a decline of just 5–7% is considered of clinical concern and would trigger clinical management. The TDF-ART association with BMD was not observed to differ across four countries spread across Southern, Central and East Africa which increases the generalizability of our findings. Sub-study participants' characteristics other than country did not differ substantially at baseline from PROMISE Postpartum component women who did not co-enroll. Likewise, the standardized DXA procedures and blinded centralized readings contributed to robustness and validity of the data. Most notably, maternal ART was initiated following randomization of healthy participants as opposed to following a clinical management decision, alleviating the potential bias present in previous publications of other observational cohorts, alluded to in Bedimo's systematic review [44].

A relative study limitation is that 15% of participants were missing data for the primary outcome measure due to not completing follow-up or lacking a week 74 DXA scan. However, frequency and reasons did not differ by study arm and the sample size achieved was able to demonstrate a difference between study arms of 2–3%. We also identifed a number of notable limitations of the study design. Firstly, we assumed breastfeeding cessation would take place 12–15 months postpartum and timed the week 74 DXA scan to occur after BMD restitution was expected to have taken place, but only half of the women had done so in our study. However, the sensitivity analysis performed suggests that the treatment effect of TDF-ART on BMD remained in women who had ceased to breast feed by week 74. Secondly, the ART regimen used in PROMISE was Protease Inhibitor-based TDF-ART. Not having a comparison ART regimen prevents evaluation of the relative roles of each agent in the observed BMD changes and restricts our ability to project what may be expected when TDF is combined with other agents. Nevertheless, TDF use is widespread for HIV treatment and prevention and PI-based TDF-ART remains second-line therapy and in use by large numbers of women globally. Ongoing studies will document the effect on maternal BMD of newer ARVs used for HIV treatment and prevention. In conclusion, this study among HIV-infected breastfeeding women in resource limited settings, demonstrated greater decrease in maternal bone mineral density up to 18 months postpartum among women randomized to TDF-ART compared to infants who were randomized to daily nevirapine prophylaxis during breastfeeding. While this difference may not be of immediate clinical significance, further study is necessary to adequately address whether there is full or only partial recovery of BMD following cessation of breastfeeding and the longer term impact through menopause of repeat pregnancies and extended periods of breastfeeding among women with HIV on life-long ART in resource limited settings. The findings also highlight the need for bone metabolism studies to further elucidate the underlying biologic patterns and mechanisms of bone loss and recovery among mothers on life-long TDF-ART.

## Supporting information

**S1 Fig. Maternal randomization schema during the PROMISE study for mothers enrolled in the postpartum TDF exposure component of P1084s.** Under version 2.0 of the trial

protocol only women who were positive for hepatitis B surface antigen were randomly assigned to tenofovir-based antiretroviral therapy (ART); under (last) version 3.0, all women could be assigned to any of the three regimens (1). P1084s postpartum enrolled completely under PROMISE protocol version 2.0. After delivery, women were randomized to continue or initiate ART or to no ART (2). In a third randomization (not shown) after weaning, eligible women were randomized again to continue ART or to a no ART arm (3). Those who had been previously randomized to no ART were continued on no ART. The first antepartum randomization was Apr. 11, 2011. After the results of the START study in July 2015 (4), all women were recommended to initiate ART. At that time, all P1084s women who enrolled in the Postpartum Exposure Component had discontinued or completed the 74 week follow-up for this substudy. All women who met specific treatment guidelines and all infants who were confirmed to be infected with the human immunodeficiency virus (HIV) began ART immediately. The above schema represents breast-feeding women. AP = Antepartum, IP = intrapartum, PP = postpartum. TDF = tenofovir disoproxil fumarate, FTC = emtricitabine, ZDV = zidovudine, sdNVP = single dose nevirapine. (TIF)

**S1 Table. Baseline characteristics by P1084s Co-enrollment.**
(DOCX)

**S1 File.**
(PDF)

**S1 Checklist.**
(PDF)

## Acknowledgments

The PROMISE study team gratefully acknowledges the dedication and commitment of the mother-infant pairs without whom this study would not have been possible. We acknowledge the research teams at each of the participating sites: **PROMISE Study Team Members**: Judith Currier, Katherine Luzuriaga, Adriana Weinberg, James McIntyre, Tsungai Chipato, Karin Klingman, Renee Browning, Mireille Mpoudi-Ngole, Jennifer S. Read, George Siberry, Heather Watts, Lynette Purdue, Terrence Fenton, Linda Barlow-Mosha, Mary Pat Toye, Mark Mirochnick, William B. Kabat, Benjamin Chi, Marc Lallemant, Karin Nielsen; Statistical and Data Analysis Center, Harvard T.H. Chan School of Public Health: Kevin Butler MS, Konstantia Angelidou MS, David Shapiro, and Sean Brummel. IMPAACT Operations Center: Anne Coletti, Veronica Toone, Megan Valentine, Kathleen George; Frontier Science Data Management Center: Amanda Zadzilka, Michael Basar, Amy Jennings, Adam Manzella.

 **INDIA**. Sandesh Patil; Ramesh Bhosale; Neetal Nevreka

 **MALAWI**. Blantyre: Salome Kunje; Alex Siyasiya, Certificate in Microbiology; Mervis Maulidi. Lilongwe/UNC: Francis Martinson; Ezylia Makina; Beteniko Milala.

 **SOUTH AFRICA**. Durban Paediatric: Nozibusiso Rejoice Skosana; Sajeeda Mawlana, MBChB Family Clinical Research Unit: Jeanne Louw; Magdel Rossouw; Lindie Rossouw. Shandukani Research: Masebole Masenya; Janet Grab. Soweto: Nasreen Abrahams; Mandisa Nyati; Sylvia Dittmer. Umlazi CRS: Dhayendre Moodley; Vani Chetty; Alicia Catherine Desmond.

 **TANZANIA**. Kilimanjaro Christian Medical Centre: Boniface Njau; Cynthia Asiyo; Pendo Mlay.

 **UGANDA.** MU-JHU Research Collaboration: Maxensia Owor; Moreen Kamateeka; Dorothy Sebikari.

ZAMBIA. George Clinic: Felistas M. Mbewe; Martin Mwalukanga.

ZIMBABWE. Harare Family Care: Tichaona Vhembo; Nyasha Mufukari. Seke North: Lynda Stranix-Chibanda; Teacler Nematadzira; Gift Chareka. St. Mary's: Jean Dimairo; Tsungai Chipato; Bangani Kusakara; Mercy Mutambanengwe; Emmie Marote.

## Author Contributions

**Conceptualization:** Lynda Stranix-Chibanda, Kathleen George, Lynne M. Mofenson, Mary Glenn Fowler, George K. Siberry.

**Data curation:** Lynda Stranix-Chibanda, Camlin Tierney, Bo Fan, Markus J. Sommer, John Shepherd, Mary Glenn Fowler.

**Formal analysis:** Lynda Stranix-Chibanda, Camlin Tierney, Bo Fan, Markus J. Sommer, John Shepherd, Bryan Nelson.

**Funding acquisition:** Mary Glenn Fowler, George K. Siberry.

**Investigation:** Lynda Stranix-Chibanda, Dorothy Sebikari, Jim Aizire, Sufia Dadabhai, Admire Zanga, Cynthia Mukwasi-Kahari, Tichaona Vhembo, Avy Violari, Gerard Theron, Dhayandre Moodley, Bo Fan, Markus J. Sommer, John Shepherd.

**Methodology:** Lynda Stranix-Chibanda, George K. Siberry.

**Project administration:** Lynda Stranix-Chibanda, Kathleen George.

**Supervision:** Lynda Stranix-Chibanda, Mary Glenn Fowler, George K. Siberry.

**Writing – original draft:** Lynda Stranix-Chibanda, Camlin Tierney, Bryan Nelson, Mary Glenn Fowler, George K. Siberry.

**Writing – review & editing:** Dorothy Sebikari, Jim Aizire, Sufia Dadabhai, Admire Zanga, Cynthia Mukwasi-Kahari, Tichaona Vhembo, Avy Violari, Gerard Theron, Dhayandre Moodley, Kathleen George, Bo Fan, Markus J. Sommer, Renee Browning, Lynne M. Mofenson, John Shepherd.

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
