## [Decision Letter · Decision Letter 0]

3 Nov 2020

PONE-D-20-15228

Impact of postpartum tenofovir-based antiretroviral therapy on bone mineral density in breastfeeding women with HIV enrolled in a randomized clinical trial

PLOS ONE

Dear Dr. Stranix-Chibanda,

Thank you for submitting your manuscript to PLOS ONE. After careful consideration, we feel that it has merit but does not fully meet PLOS ONE’s publication criteria as it currently stands. Therefore, we invite you to submit a revised version of the manuscript that addresses the points raised during the review process.

Please update the Results and analyses as suggested by the reviewers.

We look forward to receiving your revised manuscript.

Kind regards,

Nancy Beam, PhD

Staff Editor

PLOS ONE

Journal Requirements:

2.Thank you for including your ethics statement:

"The sub-study was approved by an institutional review board or ethics committee at each of the multiple sites and corresponding collaborating institutions in the United States. Written informed consent was obtained prior to study participation."

3.We note that you have indicated that data from this study are available upon request. PLOS only allows data to be available upon request if there are legal or ethical restrictions on sharing data publicly. For information on unacceptable data access restrictions, please see http://journals.plos.org/plosone/s/data-availability#loc-unacceptable-data-access-restrictions.

4. One of the noted authors is a group or consortium [PROMISE P1084s study team]. In addition to naming the author group, please list the individual authors and affiliations within this group in the acknowledgments section of your manuscript. Please also indicate clearly a lead author for this group along with a contact email address.

5.Thank you for stating the following in the Financial Disclosure section:

[Overall support for the International Maternal Pediatric Adolescent AIDS Clinical Trials Network (IMPAACT) was provided by the National Institute of Allergy and Infectious Diseases (NIAID) of the National Institutes of Health (NIH) under Award Numbers UM1AI068632 (IMPAACT LOC), UM1AI068616 (IMPAACT SDMC) and UM1AI106716 (IMPAACT LC), with co-funding from the Eunice Kennedy Shriver National Institute of Child Health and Human Development (NICHD) and the National Institute of Mental Health (NIMH). The content is solely the responsibility of the authors and does not necessarily represent the official views of the NIH. Antiretrovirals were provided free of charge for the PROMISE P1084s study by AbbVie, Gilead Sciences, and GlaxoSmithKline. Funding to purchase bone density scanners was provided for the PROMISE P1084s study by Gilead Sciences. The funders had no role in study design, data collection and analysis, decision to publish, or preparation of the manuscript.]. 

We note that you received funding from a commercial source: [Gilead Sciences]

Reviewers' comments:

Reviewer's Responses to Questions

**Comments to the Author**

1. Is the manuscript technically sound, and do the data support the conclusions?

Reviewer #1: Partly

Reviewer #2: Partly

2. Has the statistical analysis been performed appropriately and rigorously? 

Reviewer #1: No

Reviewer #2: Yes

3. Have the authors made all data underlying the findings in their manuscript fully available?

Reviewer #1: Yes

Reviewer #2: Yes

4. Is the manuscript presented in an intelligible fashion and written in standard English?

Reviewer #1: Yes

Reviewer #2: Yes

5. Review Comments to the Author

Reviewer #1: This is a well conducted study nested within the PROMISE randomised controlled trial conducted in four African countries. Selection, confounding bias and bias due to drop-out seems to be small. Also, the dilution of the effect due to people initially allocated to control and subsequently initiating TDF appears to be a conservative bias (might be worth mentioning in the Discussion).

However, I have a number of points that need to be addressed, especially regarding the statistical analysis.

Main Points

1. Line 109. The authors state that the main objective was to evaluate the possible impact of TDF exposure on BMD when used in combination with extended breastfeeding beyond the first year. In practice, 50% of the women in the study interrupted breastfeeding by week 61 and there were on average 17 weeks in which there was no extended breastfeeding before week 74. I believe that objectives need to be reworded accordingly or acknowledged that the data collected were shortfall for addressing the main aim.

2. Unclear why the earlier time-points (week 6 and week 26) have been ignored completely. It could be important to show whether the difference in BMD percentage change could be detected earlier on as it would re-inforce the hypothesis that breastfeeding was an issue. The most likely explanation for the findings seems to be the cumulative exposure/exacerbation of the effect of TDF in co-presence of boosted PIs instead.

3. Lines 156- 158. Were technicians evaluating the scans blind to treatment allocation? How can authors exclude that oucomes were not affected by technicians knowing who had received TDF?

4. Were percentage changes distributed approximately normal? It is unusual to use t-test and linear regression for a percentage ranging between -15% to +15%. Because confounding does not seem to be a major issue, how would the results look when using non-parametric tests? Why not using absolute change after controlling for baseline level?

5. Confounding, mediation, interaction. There is a lot of confusion on these issues. I think that it is agreeable that AP randomization, country, age, baseline weight, parity at PROMISE entry and plasma HIV-1RNA level are potential confounders. There is some evidence that AP randomization and baseline weight are effect modifiers for the primary endpoint while age and baseline weight for the secondary endpoint (Table 2). Therefore, these should be removed as confounders from the two analyses, respectively. Regarding the values measured at week 74, none of these should be included as confounders or as stratification factors in the subgroup analyses: breastfeeding status and current receipt of TDF are time-dependent confounders, so a standard regression analysis is unable to control for the possible introduced bias. Current weight, time and use of contraceptive are even potential mediators in the causa pathway between TDF and BMD change so should not be controlled for and cannot be evaluated as possible effect modifiers. On top of this, there should be consistency between text (Methods lines 176-178 vs. Discussion, lines 286-287) and the footnote of Figure 1 and what shown in Table 2 regarding the various adjustments.

6. If authors are really worried about drop-outs they should perform a per-protocol analysis controlling for cross-over dilution and possible informative censoring.

7. What is a change in BMD that can be considered as clinically relevant? In line 320 this is set to be as 5-7%. The actual estimate from the analysis in in the range of 2-3%, the long-term clinical implication of which are said to be unknown (lines 289-291). If this is the case than the study only shows a statistical difference which has unclear clinical implications and contradicts the conclusions.

8. Because INSTI have replaced PI/r even in resource limited settings, will TDF have a similar impact on BMD in the future?

Other points

1. Line 139. A P is missing (P1084 sub-study)

2. Lines 180-181. Most statisticians are moving away for the concept of a threshold for significance such as the arbitrary 0.05 cut-off. Best to speak more generally about compatibility of the data with the null hypothesis.

3. Lines 321-322. Despite the lack of evidence for an interaction (the study was not powered to detect these), percent BMD change in Malawi appeared to be much smaller than that seen in other countries for the primary endpoint (Table 2).

4. Table 2. Typically this is shown as a forest plot, instead of a Table, although this mainly pertains to style

5. Figure 2. Unusual to show box-plots (medians, IQR) when the assumption is that the distribution of a percentage change is approximately normal so that means and SD are shown and means compared (see main point 4 above).

Reviewer #2: This paper evaluated the impact of postpartum tenofovir-based antiretroviral therapy on bone mineral density in breastfeeding women with HIV enrolled in the IMPAACT P1084s sub-study of the PROMISE randomized trial. In general, pregnancy itself may lead to bone loss but if followed by lactation, it will have protective effect on bone density while the duration of lactation and parity may modulate its effect. In this study, having being exposed to TDF-ART during 74-week postpartum breastfeeding led to significant declines (-2.5 to 3 percent lower) than having being exposed to nevirapine alone.

Specific comments

1. Although randomization and number of participants may have overcome bias, no data on diet or physical activity, calcium and vitamin D status or supplements are given.

2. More importantly than knowing the percent decrease in bone mineral density would have been which the clinical impact was: changes from normal to osteopaenia or from osteopaenia to osteoporosis, and incident bone fractures.

6. PLOS authors have the option to publish the peer review history of their article (what does this mean?). If published, this will include your full peer review and any attached files.

Reviewer #1: No

Reviewer #2: **Yes: **Esteban Martinez

---

## [Author Response · Author response to Decision Letter 0]

9 Dec 2020

Reviewer #1—

General comment

This is a well conducted study nested within the PROMISE randomised controlled trial conducted in four African countries. Selection, confounding bias and bias due to drop-out seems to be small. Also, the dilution of the effect due to people initially allocated to control and subsequently initiating TDF appears to be a conservative bias (might be worth mentioning in the Discussion). However, I have a number of points that need to be addressed, especially regarding the statistical analysis.

General Response:

Thank you. We have taken note of the points raised.

Main Points

M1.1. Line 109. The authors state that the main objective was to evaluate the possible impact of TDF exposure on BMD when used in combination with extended breastfeeding beyond the first year. In practice, 50% of the women in the study interrupted breastfeeding by week 61 and there were on average 17 weeks in which there was no extended breastfeeding before week 74. I believe that objectives need to be reworded accordingly or acknowledged that the data collected were shortfall for addressing the main aim.

M1.1. Response:

Thank you for this valid comment. To clarify the main aim, the study was not designed to compare BMD changes between study arms during the period of breastfeeding. We assumed breastfeeding cessation would take place 12-15 months postpartum and timed the week 74 DXA scan to occur after BMD restitution was expected to have taken place and before most women enter a subsequent pregnancy-breastfeeding cycle. If a greater BMD decline in women on TDF-ART was documented at that time, the study team considered that could be potentially detrimental to the mother’s future bone health. The last paragraph of the Introduction has been reworded in lines 99-118 in the version with tracked changes.

M1.2. Unclear why the earlier time-points (week 6 and week 26) have been ignored completely. It could be important to show whether the difference in BMD percentage change could be detected earlier on as it would reinforce the hypothesis that breastfeeding was an issue. The most likely explanation for the findings seems to be the cumulative exposure/exacerbation of the effect of TDF in co-presence of boosted PIs instead.

M1.2 Response: 

This comment relates to the unclear phrasing of the study aim alluded to earlier in comment M1.1. BMD was not assessed at postpartum weeks 6 and 26. The revised aim in the Introduction clarifies that the findings reflect the effect of exposure to TDF-ART on BMD change between baseline and postpartum week 74 in a population of women with HIV who breastfed. 

M1.3. Lines 156-158. Were technicians evaluating the scans blind to treatment allocation? How can authors exclude that outcomes were not affected by technicians knowing who had received TDF?

M1.3 Response:

Yes—DXA operators conducting the scans and the central DXA reading unit were blinded to treatment allocation. This important point has been added to line 164-165 in the revised manuscript with changes highlighted.

M1.4. Were percentage changes distributed approximately normal? It is unusual to use t-test and linear regression for a percentage ranging between -15% to +15%. Because confounding does not seem to be a major issue, how would the results look when using non-parametric tests? Why not using absolute change after controlling for baseline level?

M1.4 Response: 

We thank the reviewer for this comment relating to the statistical analysis plan adopted by the protocol team. The t-test is relatively robust to non-normality (e.g., see Lumley, et al. Annu. Rev. Public Health, 23:151–169, 2002). The sample size of 339 (334) participants is large enough to allow the statistical effects of averaging to come into play to perform an approximate t-test with non-normal distributions. Visual inspection of the data indicates symmetric and approximate normal distribution. Additionally, percent change in BMD has commonly been used as the outcome measure in these younger patients where standardized outcome data are lacking. Many experts regard a change or difference in BMD of 3-5% (or 0.5 SD) as significant in whom standardized outcome data are lacking. Results are similar when using a non-parametric Wilcoxon rank sum test (p<.001)

M1.5. Confounding, mediation, interaction. There is a lot of confusion on these issues. I think that it is agreeable that AP randomization, country, age, baseline weight, parity at PROMISE entry and plasma HIV-1RNA level are potential confounders. There is some evidence that AP randomization and baseline weight are effect modifiers for the primary endpoint while age and baseline weight for the secondary endpoint (Table 2). Therefore, these should be removed as confounders from the two analyses, respectively. Regarding the values measured at week 74, none of these should be included as confounders or as stratification factors in the subgroup analyses: breastfeeding status and current receipt of TDF are time-dependent confounders, so a standard regression analysis is unable to control for the possible introduced bias. Current weight, time and use of contraceptive are even potential mediators in the causal pathway between TDF and BMD change so should not be controlled for and cannot be evaluated as possible effect modifiers. On top of this, there should be consistency between text (Methods lines 176-178 vs. Discussion, lines 286-287) and the footnote of Figure 1 and what shown in Table 2 regarding the various adjustments.

M1.5 Response:

The study team has taken your comments into consideration and removed post-randomization covariates from Table 2 and Figure 2. We removed also the main effects baseline adjusted analysis from Figure 2. Please see the new Tables and Figures submitted. The inconsistent text between Methods and Discussion has been reconciled by removing the factors listed in the Discussion, reflected in the first two highlighted paragraphs in the manuscript. 

M1.6. If authors are really worried about drop-outs they should perform a per-protocol analysis controlling for cross-over dilution and possible informative censoring.

M1.6 Response:

Subgroup results for lumbar spine BMD among participants who remained on their assigned regimen, and who ceased breastfeeding from Table 2 were moved to the Results section line 318-323 in the highlighted manuscript. Also, we thank this reviewer for pointing out that the dilution of the effect due to people initially allocated to control and subsequently initiating TDF appears to be a conservative bias and have added this point to the Discussion lines 330-331 in the highlighted manuscript. 

M1.7. What is a change in BMD that can be considered as clinically relevant? In line 320 this is set to be as 5-7%. The actual estimate from the analysis in in the range of 2-3%, the long-term clinical implication of which are said to be unknown (lines 289-291). If this is the case than the study only shows a statistical difference which has unclear clinical implications and contradicts the conclusions.

M1.7 Response:

The reviewer is correct; we do not think that the difference had an immediate clinical implication and the long-term effect remains unknown. The conclusion has been reworded to make that clear in lines 341-349 in the highlighted manuscript. 

M1.8. Because INSTI have replaced PI/r even in resource limited settings, will TDF have a similar impact on BMD in the future?

M1.8 Response:

That remains to be seen. Our findings can’t be used to speculate the effect on BMD when TDF is combined with other ARV agents. This has been clarified in the revised Limitations section of the Discussion in lines 389-396.

Other points

O1.1. Line 139. A P is missing (P1084 sub-study).

O1.1 Response:

Corrected. Thank you for pointing out this typographical error which is rectified in the highlighted manuscript.

O1.2. Lines 180-181. Most statisticians are moving away for the concept of a threshold for significance such as the arbitrary 0.05 cut-off. Best to speak more generally about compatibility of the data with the null hypothesis.

O1.2 Response: 

Since our analysis pre-specified a 0.05 significance level, we have left this statement in the manuscript. However, we reviewed and revised statements in the text about significance in the Results and Discussion sections and, for example, the Conclusion was revised by deleting the word significantly; ‘significantly greater’.

O1.3. Lines 321-322. Despite the lack of evidence for an interaction (the study was not powered to detect these), percent BMD change in Malawi appeared to be much smaller than that seen in other countries for the primary endpoint (Table 2).

O1.3 Response:

We note this comment and inserted this observation in Results section lines 310-312 in the highlighted manuscript. 

O1.4. Table 2. Typically this is shown as a forest plot, instead of a Table, although this mainly pertains to style

O1.4 Response

Thank you for that suggestion which we have accepted. Table 2 has been replaced with forest plots (Figure 3A and 3B).

01.5. Figure 2. Unusual to show box-plots (medians, IQR) when the assumption is that the distribution of a percentage change is approximately normal so that means and SD are shown and means compared (see main point 4 above).

O1.5 Response

Thank you for your comment. We presented the boxplots for descriptive purposes. To provide more information on the distribution (see Comment M1.4) we have substituted the box plots with violin plots. These violin plots present density, quartiles, median, mean with 95% CI whiskers, and individual data points (please see revised Figure 2).

Reviewer #2— 

General Comment 

This paper evaluated the impact of postpartum tenofovir-based antiretroviral therapy on bone mineral density in breastfeeding women with HIV enrolled in the IMPAACT P1084s sub-study of the PROMISE randomized trial. In general, pregnancy itself may lead to bone loss but if followed by lactation, it will have protective effect on bone density while the duration of lactation and parity may modulate its effect. In this study, having being exposed to TDF-ART during 74-week postpartum breastfeeding led to significant declines (-2.5 to 3 percent lower) than having being exposed to nevirapine alone.

General response:

No response needed to this summary from the reviewer.

Specific comments

2.1. Although randomization and number of participants may have overcome bias, no data on diet or physical activity, calcium and vitamin D status or supplements are given.

2.2 Response:

Calcium status is added to the baseline characteristics in Table 1. Data on diet, physical activity and Vitamin D status are not available at this time. The protocol did not specify supplements to be given, inserted to the Methods section line 151 in the highlighted manuscript.

2.2. More importantly than knowing the percent decrease in bone mineral density would have been which the clinical impact was: changes from normal to osteopaenia or from osteopaenia to osteoporosis, and incident bone fractures.

2.2 Response:

The reviewer is correct; ultimately it is the clinical implications of bone loss that are important. To respond to this comment, incident fractures through exit from the PROMISE main study are summarized for each study arm in the Results section in lines 273-275 in the highlighted manuscript. Standardized outcome data on this population of young women who breastfeed are lacking, hence the clinical relevance of meeting traditional thresholds for osteopaenia or osteoporosis is unclear. The period of follow-up in this study was relatively short to document the full effect of TDF-ART on maternal bone health, especially the implications on bone loss following menopause. However, our data contribute towards understanding its role and remain a valuable addition to knowledge. The Conclusion has been revised to state what research questions remain unanswered in lines 400-407.

---

## [Editor Report · Decision Letter 1]

18 Jan 2021

Impact of postpartum tenofovir-based antiretroviral therapy on bone mineral density in breastfeeding women with HIV enrolled in a randomized clinical trial

PONE-D-20-15228R1

Dear Dr. Stranix-Chibanda,

We’re pleased to inform you that your manuscript has been judged scientifically suitable for publication and will be formally accepted for publication once it meets all outstanding technical requirements.

Kind regards,

Esteban Martinez

Guest Editor

PLOS ONE

Additional Editor Comments (optional):

I declare that I participated as a reviewer for the initial evaluation of this manuscript.

Authors have adequately addressed reviewers' comments and have provided satisfactory responses to queries.

The manuscript has been definitely improved and is worth to publish now.
---

## [Editor Report · Acceptance letter]

22 Jan 2021

PONE-D-20-15228R1 

Impact of postpartum tenofovir-based antiretroviral therapy on bone mineral density in breastfeeding women with HIV enrolled in a randomized clinical trial 

Dear Dr. Stranix-Chibanda:

I'm pleased to inform you that your manuscript has been deemed suitable for publication in PLOS ONE. Congratulations! Your manuscript is now with our production department. 

Kind regards, 

on behalf of

Professor Esteban Martinez 

Guest Editor

PLOS ONE